# Dissecting Total Plasma and Protein-Specific Glycosylation Profiles in Congenital Disorders of Glycosylation

**DOI:** 10.3390/ijms21207635

**Published:** 2020-10-15

**Authors:** Agnes L. Hipgrave Ederveen, Noortje de Haan, Melissa Baerenfaenger, Dirk J. Lefeber, Manfred Wuhrer

**Affiliations:** 1Center for Proteomics and Metabolomics, Leiden University Medical Center, Albinusdreef 2, 2333 ZA Leiden, The Netherlands; A.L.Hipgrave_Ederveen@lumc.nl (A.L.H.E.); N.de_Haan@lumc.nl (N.d.H.); 2Department of Neurology, Donders Institute for Brain, Cognition and Behavior, Radboud University Medical Centre, Geert Grooteplein 10, 6525 GA Nijmegen, The Netherlands; Melissa.Barenfanger@radboudumc.nl; 3Department of Laboratory Medicine, Translational Metabolic Laboratory, Radboud University Medical Center, Geert Grooteplein 10, 6525 GA Nijmegen, The Netherlands

**Keywords:** congenital disorders of glycosylation, glycomics, sialic acid linkage isomers, mass spectrometry

## Abstract

Protein *N*-glycosylation is a multifactorial process involved in many biological processes. A broad range of congenital disorders of glycosylation (CDGs) have been described that feature defects in protein *N*-glycan biosynthesis. Here, we present insights into the disrupted *N*-glycosylation of various CDG patients exhibiting defects in the transport of nucleotide sugars, Golgi glycosylation or Golgi trafficking. We studied enzymatically released *N*-glycans of total plasma proteins and affinity purified immunoglobulin G (IgG) from patients and healthy controls using mass spectrometry (MS). The applied method allowed the differentiation of sialic acid linkage isomers via their derivatization. Furthermore, protein-specific glycan profiles were quantified for transferrin and IgG Fc using electrospray ionization MS of intact proteins and glycopeptides, respectively. Next to the previously described glycomic effects, we report unprecedented sialic linkage-specific effects. Defects in proteins involved in Golgi trafficking (COG5-CDG) and CMP-sialic acid transport (SLC35A1-CDG) resulted in lower levels of sialylated structures on plasma proteins as compared to healthy controls. Findings for these specific CDGs include a more pronounced effect for α2,3-sialylation than for α2,6-sialylation. The diverse abnormalities in glycomic features described in this study reflect the broad range of biological mechanisms that influence protein glycosylation.

## 1. Introduction

Glycosylation is an important co- and post-translational modification that modulates protein function [1,2]. This abundant protein modification is highly complex as many different factors influence the extent of variation in glycan structure, branching and elongation. Important regulators in the biosynthesis of protein glycans are the local availability of nucleotide sugars and glycan modifying enzymes, as well as the protein turn-over time in the endoplasmic reticulum and Golgi apparatus [3,4]. The spectrum of protein glycoforms of an individual is influenced by genetic as well as environmental factors. Congenital disorders of glycosylation (CDGs) are characterized by defective synthesis causing aberrant glycan structures due to specific genetic defects in various steps along the glycan biosynthetic pathways for protein *N*- and *O*-glycosylation, glycosylphosphatidylinositol, and lipid glycosylation [5]. 

CDGs are rare and complex to diagnose as case severity and disease symptoms can vary. Conventional clinical diagnostic techniques include screening for glycosylation defects by transferrin isoelectric focussing (TIEF) [6,7]. Recent advances in the field include the implementation of mass spectrometric analysis of the abnormal glycan structures attached to transferrin (Tf) to support diagnosis [8,9,10]. Specific Tf glycosylation patterns driven by (abnormal) glycan structure abundances often correlate with very specific defects of e.g., nucleotide sugars or enzymes in the biosynthetic pathway. While the inclusion of Tf glycosylation analysis in clinical routine has been crucial to decipher many of the unresolved cases of CDG [11], this protein only displays a limited set of *N*-glycans and mainly reflects the physiological status of the liver. For some CDGs, such as MOGS-CDG, transferrin glycosylation was shown to be normal, while IgG glycosylation showed characteristic glycosylation abnormalities [12].

Expanding mass spectrometry (MS)-based techniques for the investigation of CDGs to a variety of glycoproteins present in a biological sample will provide information on a broader collection of glycan structures and related defects. Therefore, in this study, we selected two *N*-glycosylated proteins to study a range of CDGs, namely the plasma B cell derived immunoglobin G (IgG) and liver synthetized Tf, and explored an array of glycosylation features from total plasma proteins (total plasma *N*-glycome, TPNG). Importantly, structural information on sialic acid linkages was obtained. We employed matrix-assisted laser desorption/ionization time-of-flight MS (MALDI-TOF-MS) for TPNG and IgG after enzymatic release of *N*-glycans and linkage-specific sialic acid derivatization. This derivatization approach allowed the facile distinction between α2,3- and α2,6-linked sialic acids. As a known biochemical marker for CDG, glycosylation patterns of intact transferrin were analyzed by the established CDG diagnostic ESI-MS method [9,13]. Furthermore, we applied subclass-specific IgG Fc *N*-glycosylation analysis by nanoscale liquid chromatography (LC)-MS of tryptic glycopeptides. These methods were applied to plasma samples from healthy individuals and various CDG patients exhibiting deviations in *N*-glycan biosynthesis caused by genetic defects in nucleotide sugar transport, Golgi glycosylation or Golgi trafficking, providing insights into the associated glycomic changes.

## 2. Results

### 2.1. Glycosylation Analysis of Plasma Proteins in Controls

In order to obtain insights into the disrupted glycosylation phenotype of CDGs, we applied a range of MS glycomics methods to plasma samples of 20 individual CDG patients and 10 healthy controls (Table 1 and Appendix A). Glycan structures were deduced on the basis of tandem MS data (Appendix A) as well as literature knowledge of *N*-glycan structures and biosynthesis [14,15,16,17,18] (Figure 1). Within the glycomics data, the abundances of glycan structural features (derived traits) were calculated for TPNG, IgG released *N*-glycans and IgG Fc *N*-glycopeptides, giving insights into differences in *N*-glycan biosynthesis (Appendix A). 

For healthy controls, the analysis of TPNG revealed a diverse group of *N*-glycans dominated by diantennary structures carrying in part α2,6-linked sialic acid as well as tri- and tetraantennary structures in lower abundance carrying sialic acids in different linkages (Figure 2B) [20,21,22]. Additionally, minor amounts of oligomannosidic and hybrid-type *N*-glycans were detected. Studying IgG glycosylation via the site-specific analysis of Fc *N*-glycopeptides and the released *N*-glycan analysis revealed mainly diantennary complex-type *N*-linked glycans with high levels of core fucosylation, intermediate levels of galactosylation, and low levels of sialylation and bisection (Appendix A). Healthy controls showed mainly α2,6-linked sialic acids on IgG glycans [23,24]. The healthy control data were found to be in accordance with literature.

### 2.2. N-Glycosylation in Nucleotide Sugar Transporter Defects 

For the GDP-fucose transporter defect SLC35C1-CDG, intact transferrin MS profiling showed no deviating pattern (Appendix A), as transferrin naturally contains a low degree of fucosylation [9]. On the contrary, a drastic decrease in the fucosylation of IgG was observed, with only 4.1% of the complex structures carrying a fucose in IgG1, while for the healthy controls the range was 92.4–98.4% (Figure 3A). TPNG analysis of SLC35C1-CDG revealed lower levels of overall *N*-glycan fucosylation (2.6%, control range 26.5–35.6%; Appendix A). Furthermore, the α2,6 sialylation of specifically non-fucosylated A2 glycans in TPNG was lower for the SLC35C1-CDG patient, while the overall α2,6-sialylation of diantennary (A2) structures was unaffected (Figure 4A,D). 

For SLC35A1-CDG, a CMP-sialic acid transporter defect, intact transferrin MS profiling revealed lowered sialylation (Appendix A). Specifically, the main structure (H5N4S2) on intact Tf was found to be 45.9% whilst for the healthy controls the range was 79.0–89.0%. Lower levels of sialylated *N*-glycans for SLC35A1-CDG were observed in TPNG as well, including the triantennary (Figure 4B,E) and tetraantennary species (Figure 4C,F). The diantennary non-fucosylated *N*-glycans showed lower levels of both α2,3- and α2,6-sialylation (Figure 4A). Of note, the effects on α2,3-sialylation (4.1%, control range 8.4–13.9%) showed to be more pronounced than on α2,6 sialylation (53.9%, control range 62.9–68.7%).

For the UDP-galactose transporter defect (SLC35A2-CDG), intact Tf profiles showed truncated glycans lacking galactose and sialic acid (Appendix A). Similarly, released *N*-glycan data and the IgG Fc glycopeptide data showed slightly reduced levels of galactosylation of complex-type glycans (CG) (Appendix A). 

### 2.3. Defects in Golgi N-Glycosylation

B4GALT1-CDG showed a Tf profile vastly lacking galactosylation (e.g., 77,743 Da, Appendix A). TPNG and IgG glycosylation profiles likewise revealed low levels of galactosylation and an overall reduced sialylation of complex-type glycans (CS). Remarkably, the level of sialylation per galactose (A2GS) was elevated in total IgG glycans (Appendix A). This effect was not observed for the Fc specific glycopeptide data (Appendix A), suggesting that the increased sialylation efficiency supposedly occurs on the variable domain. B4GALT1-CDG showed a higher degree of bisection compared to the healthy controls in both total IgG and the Fc-specific analysis. 

MGAT2-CDG caused high levels of mono-antennary structures across all four MS datasets (Appendix A). Similarly, MAN1B1-CDG showed the elevation of hybrid-type structures in both the protein-specific and TPNG analysis. Higher levels of sialylation were found on IgG in MGAT2-CDG and MAN1B1-CDG patients (Appendix A). This was mainly driven by the increase of mono-antennary sialylation influencing the overall CS.

### 2.4. N-Glycosylation in Golgi Homeostasis and Trafficking Defects

Disorders of Golgi homeostasis, for example due to disruption of the V-ATPase proton pump or Golgi trafficking, are generally characterised by lowered sialylation and galactosylation in Tf MS profiling (Appendix A). Previously, we have reported a loss of triantennary structures and an increase of truncated structures to be associated with these disorders, after analysing permethylated plasma *N*-glycan with MALDI ion trap MS. However, these studies didn’t include differentiation of sialic acid linkage or additional plasma proteins.

For COG5-CDG, intact Tf MS profiling revealed low levels of sialylation with a relative increase in mono-sialylated structures. The TPNG from this patient showed lower levels of α2,3 sialylation on triantennary structures (e.g., *m*/*z* 2940.05 and 2894.01 with 1 and 2 α2,3-linked sialic acids, respectively; Figure 2C) when compared to the healthy control group (Figure 2B). The effect was stronger for α2,3-linked sialic acid (SA) than for α2,6-linked SA, as exemplified by sialylation per galactose on triantennary structures (Figure 2A). The lower levels of α2,3-linked SA came with reduced (antennary) fucosylation on tri- and tetraantennary structures (Figure 4E,F), resulting in a reduction of the sialyl Lewis^X^ motif. Although the overall effect on α2,6-sialylation was less pronounced, a distinctive reduction of the α2,6-sialylation level on fucosylated A2 glycans was observed (Figure 4D) as compared to the healthy controls.

Similar to COG5-CDG, patients diagnosed with TMEM199-CDG showed reduced sialylated structures and accumulation of mono-galactosylated species in intact Tf profiles. The most noticeable TPNG phenotype for TMEM199-CDG was the low level of α2,6-linked sialic acid on triantennary glycans (A3) (Figure 4B,E). The IgG1 Fc *N*-glycosylation traits for TMEM199-CDG patients were largely conserved, with the exception for P14, showing a lowered complex-type fucosylation of 79.4% (control reference values 92.4–98.4%, Figure 5A). 

Intact Tf profiles (Appendix A) for ATP6AP1-CDG and ATP6V0A2-CDG showed an elevation of the complex type structures lacking one sialic acid (e.g., 79,266 Da). The ATP6V0A2-CDG patients predominantly had lower levels of α2,6-linked sialylation on di- and triantennary structures on plasma proteins (Figure 4A,B,E) and ATP6AP1-CDG showed lower levels of α2,6-linked sialylation on triantennary structures (Figure 4B,E). On the contrary, the abundance of α2,3-linked sialylation of fucosylated A2 structures in TPNG was elevated. This was not only the case for ATP6AP1-CDG patients but also for CCDC115-CDG and VMA21-CDG (Figure 4D), indicating common effects following the disturbed Golgi homeostasis. Furthermore, higher levels of α2,6-linked sialylation were observed on fucosylated A4, seemingly at the expense of the α2,3 SA on non-fucosylated A4 structures. The IgG1 Fc levels of complex-type galactosylation (CG, Figure 5B) was reduced for both ATP6V0A2-CDG and ATP6AP1-CDG, leading to lowered levels of complex-type sialylation (CS) only in ATP6V0A2-CDG patients (Figure 5C). The TPNG profile of CCDC115-CDG revealed lower levels of α2,6 SA, especially on fucosylated A3 structures, next to the above-mentioned elevated levels of α2,3-linked sialylation of fucosylated A2 structures (Figure 4D,E).

The *VMA21* genetic variant in P18 with an XMEA myopathy phenotype did not result in abnormal CDG screening via Tf glycosylation profiles [25]. However, the level of (antenna) fucosylation within A3 and A4 structures found in TPNG was higher for P18 as compared to the healthy controls, regardless of sialylation (A3F; 50.4% and A4F; 60.9%, control ranges: 16.0–38.3% and 27.3–49.7%, respectively; Appendix A). Interestingly, these features were conserved for the diagnosed VMA21-CDG patient (P17). In Tf MS profiling the lack of one sialic acid (signal at 79,266 Da) was observed for multiple CDG subtypes, including P17. 

## 3. Discussion

### 3.1. N-Glycosylation

Transferrin glycosylation analysis is established for CDG diagnosis, either via isoelectric focusing or by intact mass analysis [9,10,26]. In this study, we performed various MS glycomics analyses, including novel linkage-specific sialylation analysis of total plasma proteins and IgG glycosylation, to a wide range of CDGs. The CDGs included direct defects in the glycosylation machinery and vesicular transport defects. Previously, serum protein *N*-glycosylation from CDG patients was studied with permethylated *N*-glycans by MALDI-TOF-MS [27]. Our study features linkage-specificity of derivatized sialic acids and provides insights into the late processing steps of glycan synthesis, namely the terminal sialic acid addition which occurs in the Golgi trans cisternae. In addition, we assessed the glycosylation of isolated proteins that are produced in different cell-types, i.e., Tf and IgG. It is well known that glycosylation of plasma proteins can be altered in a range of common diseases, the mechanisms of which mostly remain unsolved. The low number of patients in our cohort, especially the scarcity of patients sharing a specific genetic defect, precluded the identification of associations of glycan signatures with disease symptoms (see Table 1 for main clinical symptoms of the CDG patients). The strength of this study is thereby reflected in the identification and quantification of MS glycan signals that provide biological insights into how defined defects in *N*-glycan biosynthesis result in glycan structural features in human disease.

### 3.2. N-Glycosylation from Total Plasma Proteins

The addition of sialic acids as a late *N*-glycan processing event can mediate a variety of physiological and pathological processes. Changes in serum or plasma protein sialylation have been linked with cancer, autoimmune diseases and acute inflammation [28,29,30]. Our studies on genetically defined CDGs allowed to identify subtle differences in sialic acid linkages. Mutations in COG components lead to partial relocation and degradation of glycosyltransferases and other glycosylation activities, introducing alterations in glycosylation patterns [31,32]. Previously published serum *N*-glycan profiles for COG5-CDG were mainly characterized by decreased sialylation, but no differentiation was made between sialic acid linkage isomers [27,33]. The current results show that this effect resulted specifically in lower levels of α2,3-linked sialic acids, while α2,6-linked sialic acids were only affected to a minor extent. 

Additional CDGs caused by disrupted Golgi homeostasis likewise show a diverse array of altered glycosylation patterns. A defect in the complex that chaperones the assembly of the V0 domain of the vacuolar H+ ATPase (V-ATPase) is known to impair Golgi homeostasis [34]. Acidification is crucial for glycosylation and transport, and the proton pump responsible for acidification of the secretory pathway, is a subunit of the V-ATPase [35]. Under healthy conditions, the Golgi pH decreases from the cis to trans cisternae and is estimated to range from 6.7 to 6.0 [36]. Alterations in Golgi pH affect the glycosylation machinery due to mislocalized glycosyltransferase enzyme activity and impaired sorting processes [34,37,38,39]. For example, an increase of 0.2 pH unit in the Golgi luminal causes the mislocalization of sialyltransferase ST3-GalIII, leading to an alteration in terminal α2,3-sialylation of *N*-glycans [40]. Correspondingly, an elevated abundance of α2,3-linked sialylation of fucosylated A2 in TPNG was observed for CDGs involved in Golgi homeostasis (CCDC115-CDG, VMA21-CDG and ATP6AP1-CDG). The possible sialyl-Lewis^X^ motifs of the resulting glycan structures have been found indicative for the metastasis of several types of cancer [30,41,42]. 

TMEM199-CDG showed low levels of α2,6-linked sialic acid on both nonfucosylated and fucosylated triantennary glycans in TPNG. The TMEM199 protein is thought to predominantly localize in the ER-to-Golgi region [43]. Higher abundant α2,6-sialylation of triantennary and tetraantennary glycans are associated with increased age as well as BMI [44]. 

Reduction in sialic acid content for the affected individuals has been shown for SLC35A1-CDG and the more subtle CDGs affecting the Golgi, indicating that one of the terminal steps of glycan synthesis is impaired. Interestingly, the CMP-sialic acid transporter defect showed a more pronounced effect on α2,3-linked sialylation than on α2,6-linked sialylation, represented in a reduction of the sialylated fucosylated triantennary species. This indicates a potential decrease of sialyl-Lewis^X^ type glycans. Reduced sialyl-Lewis^X^ structures for SLC35A1-CDG have been described previously for the glycans found on patient plasma cells [45]. Possibly reduced sialyltransferase activity, reduced sialidase activity and/or increased sialylglycoprotein production could cause the difference in sialic acid linkage. The linkage-specific sialic acid changes observed in these CDG patients may be useful to study further to specifically diagnose CMP-sialic acid transporter defects. 

### 3.3. N-Glycosylation of Immunoglobin G

Each IgG molecule carries two glycans covalently attached to conserved glycosylation sites in the Fc region. The antibody variable domain may contain glycosylation sites as well, which is reported to occur on 15–25% of the molecules [46,47]. Differences in IgG glycosylation have been associated with various disease states and the current study on defined genetic defects allowed to get more insight into the regulation of aberrant IgG glycosylation.

Hypogalactosylation of transferrin is a reported marker of the genetic galactosylation deficiencies SLC35A2-CDG and B4GALT1-CDG [26,48], although SLC35A2-CDG has been reported to result in hypogalactosylation of transferrin *N*-glycans in only about half of the patients [49,50]. Here, we likewise observed reduced galactosylation for total plasma proteins and IgG connected to these diseases. Due to the scarce number of patients we were not able to conclude on the previously reported limited prevalence of hypogalactosylation in SLC35A2-CDG patients. Hypogalactosylation is known to associate with age and various (inflammatory) diseases [51]. The subjects studied here were young children of 1–2 years of age and while a lower galactosylation is associated with lower age in healthy individuals, the current observation is not biased by an age effect as healthy children are reported to exhibit galactose levels similar to young adults [24]. Moreover, due to the association between galactosylation and the inflammatory status [47], the more subtle galactosylation effects measured throughout the TPNG and IgG datasets of the CDGs have only very limited diagnostic potential. In contrast, several glycosyltransferase defects produced distinctive glycosylation profiles allowing for clear diagnostic markers in both TPNG and the protein-specific results, such as the indicative structures carrying terminal GlcNAc in B4GALT1-CDG patients.

Expression levels of transferases and glycosidases in the secretion pathway of antibody-producing plasma B cells determine the glycosylation profiles of antibodies. However, sialylation is thought to also depend on extracellular sialyltransferase expression in the circulation [52,53]. The higher degree of variable domain sialylation found on IgG from B4GALT1-CDG patients could in part stem from the hepatic asialoglycoprotein receptor which may selectively remove nonsialylated Fab glycoforms [54]. The hypersialylation observed on the Fc portion of the antibody for MGAT2- and MAN1B1-CDG patients, may convey anti-inflammatory properties as shown for other sialylated IgG1 variants [55]. 

The drastic decrease in antibody fucosylation for SLC35C1-CDG can contribute to increased binding affinity to the FcγRIIIa receptor, affecting the antibody dependent cellular cytotoxicity (ADCC). This Fc-dependent effector function of IgG is important for anti-viral immunity and anti-tumor therapies [56,57]. Characteristics of patients suffering from SLC35C1-CDG include recurrent infections, which involves innate immunity and is, amongst others, caused by defective neutrophil function [58,59].

### 3.4. Concluding Remarks

We focused our studies on CDGs featuring an abnormal *N*-glycosylation [60] and analyzed (protein-specific) glycosylation derived from the circulation. Therefore, the current work excludes the possibility to assess genetic defects of tissue-specifically expressed glycosyltransferases. Furthermore, gene defects exclusively affecting any other type of glycosylation than *N*-glycosylation were not covered. 

The diverse abnormalities in glycomic features described in this study reflect the broad range of biological mechanisms that influence protein glycosylation. While determining the exact cause of disruptive Golgi mechanisms remains elusive, the newly identified effects in specifically α2,3-sialylation might feature as a potential marker in future investigations of these defects. Additional follow-up studies with longitudinal (therapy) monitoring of glycomic features would be advantageous to pinpoint towards specific associations of glycan signatures with disease symptoms.

The tools used for describing glycomics features e.g., sialylation and glycosylation of total IgG comparison to the Fc region paves the way to specifically study Golgi defects for further functionality. Examples are the effects observed for α2,3-linked sialylation within COG5-CDG and SLC35A1-CDG, where one may speculate that either levels of CMP-sialic acid or differential effects on α2,3 versus α2,6 sialyltransferases result in a more dominant effect on α2,3 linked sialylation. The current methods can in the future be applied to study cell type-specific glycosylation in CDG patients [61]. Moreover, the B cell derived glycosylation features observed on protein-specific IgG, such as the possibly increased sialylation on the variable domain, may lead to further investigations into the upregulation of the asialoglycoprotein receptor in specific conditions. 

## 4. Materials and Methods 

### 4.1. Patient Samples

The patient and control samples were collected at Radboudumc in accordance with the Declaration of Helsinki and CMO approval 2019-5591 for biomarker studies in diagnostic samples of CDG patients and controls (Table 1 and Appendix A). The diagnosis of all CDG patients was genetically and biochemically confirmed in previous studies [13]. P18 has X-linked myopathy with excessive autophagy (XMEA) due to mutations in *VMA21* without abnormal CDG screening results [25]. 

### 4.2. Plasma Immunoglobin G Enrichment and Digestion

Using 10 µL CaptureSelect IgG-Fc (Hu) beads (Thermo Fisher Scientific, Breda, The Netherlands) IgG was captured from 2 µL plasma in 150 µL PBS during a 1 h incubation with continuous shaking. Non-binding components were removed by washing three times with PBS and three times with water. Enriched IgGs were subsequently eluted in 100 µL 100 mM formic acid by incubating the beads for 15 min at room temperature with agitation. The eluate was split (ratio; 1:3) before drying for 2 h in a vacuum centrifuge at 60 °C and 75% of the sample was used for *N*-glycan release (see next paragraph). The smaller aliquots samples (25% of the eluate) were dissolved in 20 μL 25 mM ammonium bicarbonate (pH 8) in 7.5% ACN with 150 ng sequence grade trypsin and incubated overnight at 37 °C. Glycopeptide analysis was performed with nanoLC-QTOF-MS as described previously [62]. The supposed glycopeptide peaks from all samples were selected for MS/MS analysis by collision-induced dissociation with nanoLC-IT-MS/MS as described elsewhere [63]. 

### 4.3. Released N-Glycans and MALDI-TOF-MS Analysis

The *N*-glycosylation of total plasma glycoproteins as well as affinity-purified IgG (75% of eluate) was analyzed by MALDI-TOF/TOF-MS/MS at the released *N*-glycan level. After enzymatic *N*-glycan release, sialic acids were stabilized by linkage-specific ethyl esterification [64,65]. Briefly, released glycans were added to derivatization reagent and incubated for 1 h at 37 °C. The derivatized glycans were enriched by cotton hydrophilic-interaction liquid chromatography (HILIC)−solid-phase extraction (SPE) as described before and eluted in 10 μL water [64,66]. MALDI-TOF(/TOF)-MS(/MS) analysis was performed on an UltrafleXtreme (Bruker Daltonics, Bremen, Germany) equipped with a Smartbeam-II laser. The enriched ethyl-esterified glycans (2 µL) were spotted on a MALDI target (MTP AnchorChip 800/384 TF; Bruker Daltonics, Bremen, Germany) together with 1 µL 5 mg/mL super-DHB in 50% ACN and 1 mM NaOH. The spots were dried by air at room temperature. Spectra were acquired with accumulation of 10.000 laser shots at a laser frequency of 1000 Hz, using a complete sample random walk with 200 shots per raster spot. Selected peaks were fragmented via laser-induced dissociation (MALDI-TOF/TOF-MS/MS).

### 4.4. Data Processing

The glycopeptides analysed by nanoLC-MS were converted to mzXML files and automated relative quantification LaCyTools (version 1.1.0-alpha) was performed [67]. Targeted peak integration was performed on externally calibrated peaks based on signals with S/N above 27 and singly to quadruply charged species were quantified. Per IgG subclass (three chromatographic clusters were defined, one per IgG subclass; IgG1, IgG4 and IgG2/3) for these clusters, sum spectra were created, and signals were integrated to include at least 95% of the theoretical isotopic pattern. The actual presence of a (glyco-)peptide was assessed based on the mass accuracy (between −20 and 20 ppm), the deviation from the theoretical isotopic pattern (IPQ; below 0.25), and the signal-to-noise, (S/N; above six). The excluded signals were re-evaluated using the following parameters: mass accuracy between −25 and 25 ppm, IPQ below 0.15, and S/N above nine. The included charge states were summed per analyte and absolute values were normalized to the total signal intensity per IgG *N*-glycosylation site.

The MALDI-TOF-MS spectra of released *N*-glycans were converted to text files prior to automated relative glycan quantification using MassyTools (version 1.0.2-alpha) [68]. Spectra were externally calibrated based on signals with a S/N above nine and targeted peak integration was performed on a manually annotated list of glycans, including at least 95% of the theoretical isotopic pattern. The actual presence of a glycan was assessed based on the mass accuracy (between −20 and 20 ppm), IPQ (below 0.25), and the S/N (above six). Analytes were included per CDG subtype when present according to curation parameters and additional visual inspection of the raw MS(/MS) data. Glycan signals were normalized to the total signal intensity. 

An in-depth analysis of compositional features for all datasets was acquired through derived traits (Appendix A; [20,69]). 

## Figures and Tables

**Figure 1 ijms-21-07635-f001:**
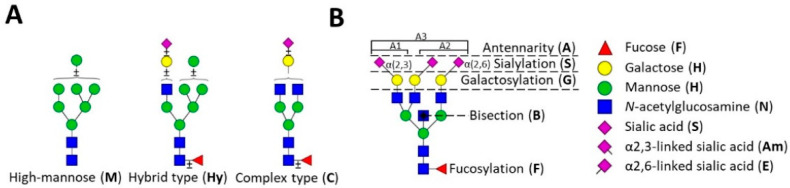
*N*-glycan structural features. The three major classes of human *N*-glycans are schematically presented (**A**) and the major structural features of complex-type *N*-glycans are introduced (**B**). Adaptation from [19].

**Figure 2 ijms-21-07635-f002:**
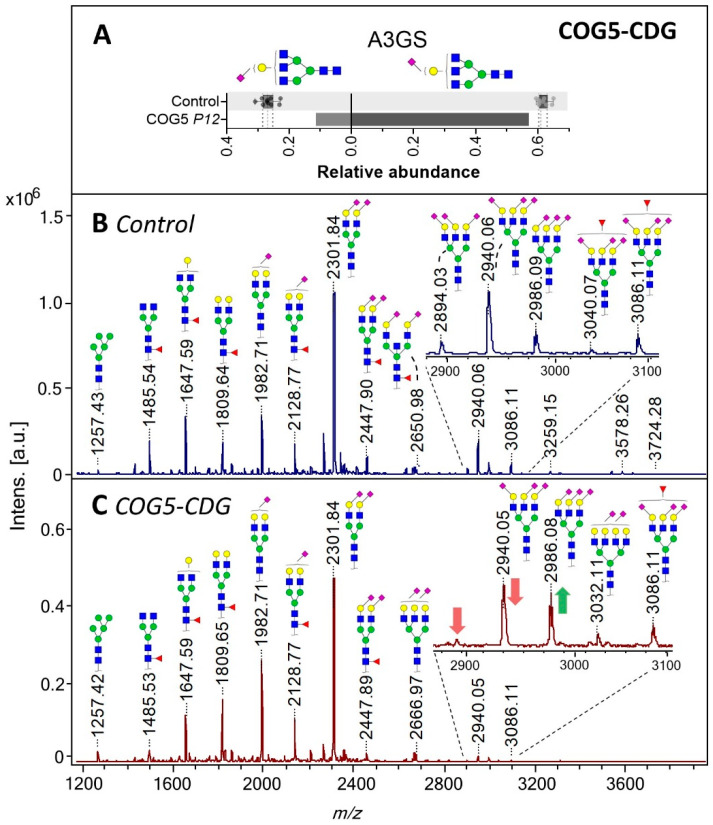
Total plasma *N*-glycome (TPNG) of a COG5-CDG patient. (**A**) Levels of α2,3-sialylation per galactose (left) and α2,6-linked sialylation per galactose (right) on triantennary glycans (A3GS) of TPNG. The box plot shows the median (dashed line) with the interquartile range (dotted lines) for healthy controls (*n* = 10) whilst the bars give the values determined for the COG5-CDG patient. (**B**) and (**C**) show the TPNG MALDI-TOF-MS profiles of healthy control 5 and the COG5-CDG patient, respectively. All glycans were detected as [M+Na]^+^. For glycan annotation see Figure 1.

**Figure 3 ijms-21-07635-f003:**
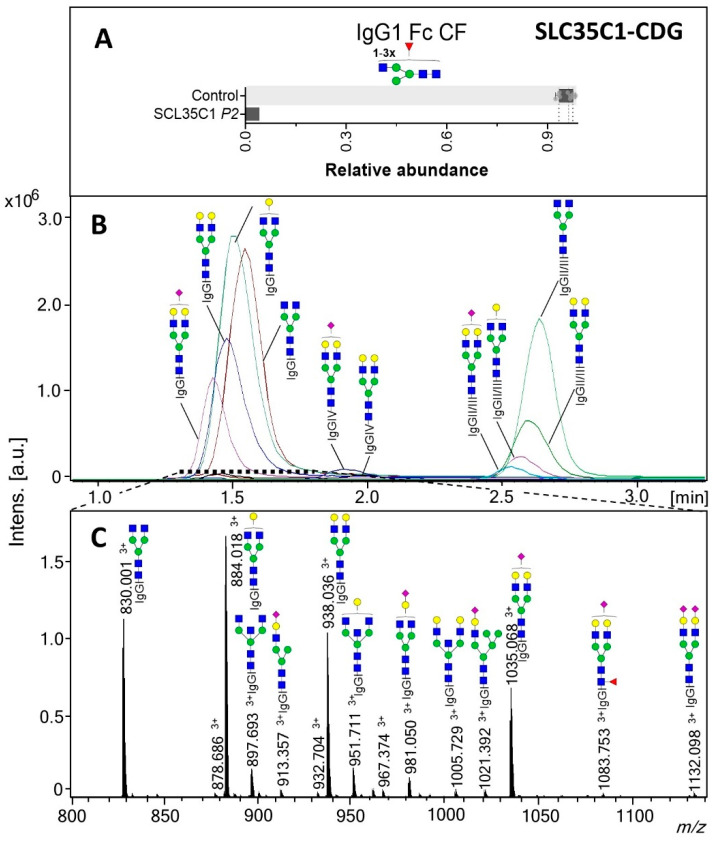
IgG1 Fc glycosylation for SLC35C1-CDG (P2). (**A**) Fucosylation (CF) of the IgG1 Fc portions. The box plot shows the median (dashed line) with the interquartile range (dotted lines) for healthy controls (*n* = 10) whilst the bar gives the value determined for the patient. (**B**) Extracted ion chromatograms (EIC) of selected glycoforms of the different IgG subclasses and (**C**) sum spectrum of the indicated window (1.3–1.8 min; IgG1 cluster) from the LC-MS analysis of the patient SLC35C1-CDG tryptic IgG Fc glycopeptides. The proposed glycopeptide structures are based on fragmentation and literature. All glycopeptide signals annotated in (**C**) were [M+3H]^3+^. For glycan structure schemes see Figure 1.

**Figure 4 ijms-21-07635-f004:**
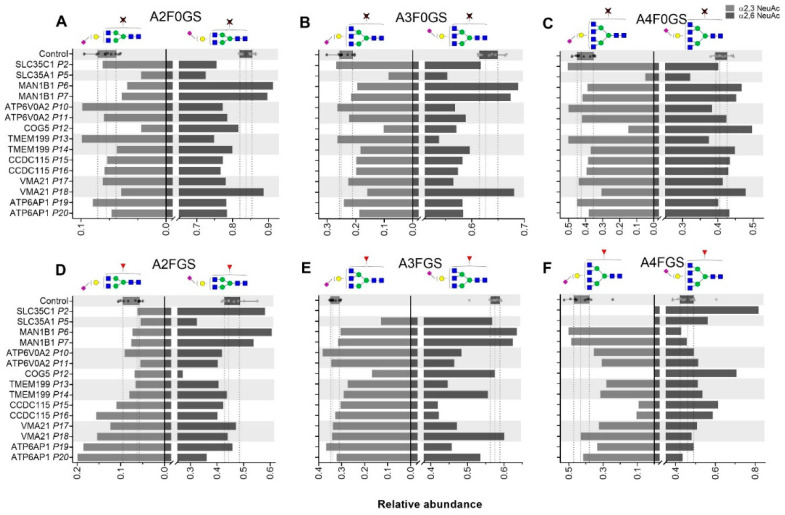
Total plasma *N*-glycome (TPNG) structural features determined for CDG patients. Levels of α2,3-sialylation per galactose (left) and α2,6-linked sialylation per galactose (right) are displayed for non-fucosylated (**A**) diantennary, (**B**) triantennary and (**C**) tetraantennary glycans as well as for fucosylated (**D**) diantennary, (**E**) triantennary and (**F**) tetraantennary glycans. The box plot shows the median (dashed line) with the interquartile range (dotted lines) for healthy controls (*n* = 10) whilst the bars give the individual values determined for the CDG patients.

**Figure 5 ijms-21-07635-f005:**
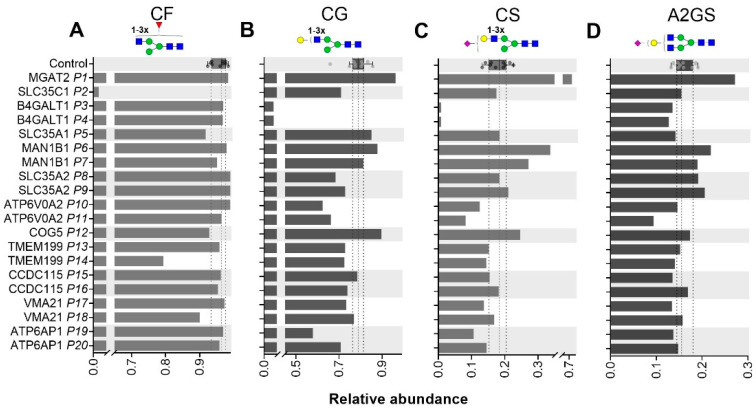
IgG1 Fc glycosylation of CDG patients. Levels of IgG1 Fc fucosylation (**A**), galactosylation (**B**) sialylation (**C**) and sialylation per galactose (**D**) as determined at the glycopeptide level are displayed for 20 CDG patients. Reference values as determined for healthy controls (*n* = 10) are shown by box plot with median (dashed line) and interquartile range (dotted lines).

**Table 1 ijms-21-07635-t001:** Characteristics of patient samples.

Patient Nr.	Gene Defect	Sex	Life Stage ^2^	Clinical Phenotype
1	*MGAT2*	- ^1^	child	Intellectual disability (ID), microcephaly, dysmorphic features, failure to thrive
2	*SLC35C1*	F	child	ID, short stature, leucocyte adhesion deficiency
3	*B4GALT1*	F	child	Mild hepatomegaly, diarrhea, dysmorphic features, abnormal coagulation
4	*B4GALT1*	M	child	
5	*SLC35A1*	F	adolescent	ID, ataxia, seizures, macrothrombocytopenia, proteinuria
6	*MAN1B1*	M	adolescent	ID, macrocephaly, truncal obesity, dysmorphic features
7	*MAN1B1*	M	adolescent	
8	*SLC35A2*	F	toddler	ID, short stature, seizures, dysmorphic features
9	*SLC35A2*	M	toddler	
10	*ATP6V0A2*	F	infant	ID, cutis laxa, dysmorphic features
11	*ATP6V0A2*	M	adolescent	
12	*COG5*	M	infant	Skeletal dysplasia, dysmorphic features, cholestasis
13	*TMEM199*	M	child	Hypercholesterolemia, elevated aminotransferases and alkaline phosphatase
14	*TMEM199*	M	adult	
15	*CCDC115*	F	child	Hypercholesterolemia, elevated aminotransferases and alkaline phosphatase, hepatomegaly, psychomotor retardation
16	*CCDC115*	M	child	
17	*VMA21*	M	adolescent	Hypercholesterolemia, elevated aminotransferases, steatosis
18	*VMA21*	M	adult	Myopathy
19	*ATP6AP1*	M	child	Jaundice, elevated aminotransferases, low immunoglobulins
20	*ATP6AP1*	M	adolescent	Jaundice, elevated aminotransferases, low immunoglobulins, hepatosplenomegaly
Healthy controls	-	6 F/4 M	3–58 (18) yrs ^3^	-

^1^ Unknown. ^2^ Infant (<1 yrs), toddler (1–3 yrs), child (3–10 yrs), adolescent (10–19 yrs) and adult (>19 yrs). ^3^ Min–Max (median).

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
