# Peer review of "Dissecting Total Plasma and Protein-Specific Glycosylation Profiles in Congenital Disorders of Glycosylation"

_ijms, 2020, doi:10.3390/ijms21207635_

Round 1
Reviewer 1 Report
The article by Hipprave Ederveen et al focuses on congenital disorders of glycosylation and characterizes the associated disrupted N-glycosylation. Using patient and control samples, N-glycans enzymatically released of total plasma proteins and affinity purified immunoglobulin G are investigated using mass spectrometry. Of particular interest, the proposed methodology highlights sialic linkage-specific effects.
The manuscript is well written and presents some very interesting works and tools that identify specific effects in sialylation patterns.
However, I have a few questions/comments that, I think, could be taken into account in order to improve the manuscript.
* Line 161: “... structures (e.g. m/z 2940.04 and 2894.03 with 1 and 2 …”
On Figure 2, I don’t see any peak at m/z 2940.04. From panel B, I read 2940.06 and from panel C, I read 2940.05.
On the different graphs presented in the article, would it be possible to materialize by an asterisk or any other sign the patients and glycosylation for which the differences with respect to the "control" group are statistically relevant ? It will make it easier for the reader to follow the presentation and discussion of the results.
Figures
* Figure 1:
On the right side of the figure, I don’t understand why Galactose and Mannose are represented by the same letter (H). Shouldn’t it be G for Galactose and M for Mannose ?
* Figure 4:
Panel A and D, on the y-axis, for homogeneity reasons, I suggest to replace SCL35C1 by SLC35C1 and VMA-21 by VMA21.
* Figure 5:
Panel A, on the y-axis, for homogeneity reasons, I suggest to replace SCL35C1 by SLC35C1, SCL35A2 by SLC35A2 and VMA-21 by VMA21.
* Supplementary figure S2:
Panel A, on the y-axis, for homogeneity reasons, I suggest to replace SCL35C1 by SLC35C1, and VMA-21 by VMA21.
Tables
* Table S3
For homogeneity reasons, I suggest to replace VMA-21 by VMA21 in columns V and W
* Table S5
For homogeneity reasons, I suggest to replace VMA-21 by VMA21 in columns V, W, AZ, BA, CD, CE, DH, DI, EL and EM
References
* reference 7 → I recommend to check the name of the journal. From pubmed, I found “Japanese journal of human genetics”
* reference 18 → initials are missing for the last authors (Radboud J. E. M. Dolhain)
* reference 19 → one initial is missing for Kammeijer (Guinevere S. M. Kammeijer)
* reference 20 → one initial is missing for two of the authors: Vreeker (Gerda C.M. Vreeker) and van der Burgt (Yuri E.M. van der Burgt)
* reference 21 → initials are missing for Tollenaar (Rob A. E. M. Tollenaar)
* reference 22 → initials are missing for three of the authors: Selman (Maurice H.J. Selman), Hazes (Johanna M.W. Hazes) and Dolhain (Radboud J.E.M. Dolhain)
* reference 35 → correct the journal reference
eLife 2017;6:e22693, doi: 10.7554/eLife.22693
* reference 43 → correct the journal reference
Biology (Basel) 2017, 6, 16, doi:10.3390/biology6010016
* reference 46 → one initial is missing for two of the authors: de Brouwer (Arjan P M de Brouwer) and Deen (Peter M T Deen)
* reference 55 → correct the name of the journal: Front Immunol instead of Frontiers in immunology. Other Frontiers papers display the "short" name
* reference 61 → one initial is missing for Koeleman (Carolien A M Koeleman)
* reference 64 → one initial is missing for the first author (Maurice H J Selman)
Author Response
The article by Hipprave Ederveen et al focuses on congenital disorders of glycosylation and characterizes the associated disrupted N-glycosylation. Using patient and control samples, N-glycans enzymatically released of total plasma proteins and affinity purified immunoglobulin G are investigated using mass spectrometry. Of particular interest, the proposed methodology highlights sialic linkage-specific effects.
The manuscript is well written and presents some very interesting works and tools that identify specific effects in sialylation patterns.
However, I have a few questions/comments that, I think, could be taken into account in order to improve the manuscript.
* Line 161: “... structures (e.g. m/z 2940.04 and 2894.03 with 1 and 2 …”
On Figure 2, I don’t see any peak at m/z 2940.04. From panel B, I read 2940.06 and from panel C, I read 2940.05.
Response:
Thank you for noticing, the correct theoretical m/z values are 2940.05 and 2894.01. The values have been adjusted in the text. The figure contains the observed m/z values obtained by mass spectrometry showing slight deviations compared to the theoretical masses described in the text. The deviations observed in Figure 2 are within the threshold described in the Materials and Methods section; <20 ppm.
On the different graphs presented in the article, would it be possible to materialize by an asterisk or any other sign the patients and glycosylation for which the differences with respect to the "control" group are statistically relevant ? It will make it easier for the reader to follow the presentation and discussion of the results.
Response:
As for most defects we only had single patients available we decided to refrain from a statistical evaluation, concluding that with these low numbers a statistical evaluation does not appear appropriate. Instead, we decided to visualize the differences between patient and control samples in the box plots. To support this visualization, we indicated the distribution of the healthy controls by highlighting the interquartile range (IQR). Patients were considered “different” from the healthy controls when their respective glycosylation trait was outside the whiskers of the box-plots (1.5xIQR) thereby being an outlier with respect to the healthy population. Obviously, follow-up studies assessing multiple samples from various patients per disease will be necessary to support a solid statistical evaluation, and to indicate the potential usefulness of the observed signatures for diagnosis or therapy monitoring.
Figures
* Figure 1:
On the right side of the figure, I don’t understand why Galactose and Mannose are represented by the same letter (H). Shouldn’t it be G for Galactose and M for Mannose ?
Response:
We have chosen for the general abbreviation H (hexose) which represents both mannose and galactose. This simplifies the compositional structural assignment of the represented glycans. Of note, in mass spectrometric analysis we are not able to distinguish between these monosaccharides, but rather assign hexoses as mannose or galactose on the basis of glycobiological knowledge and literature. The abbreviation (G) is already used to indicate galactosylation in the derived trait calculations.
* Figure 4:
Panel A and D, on the y-axis, for homogeneity reasons, I suggest to replace SCL35C1 by SLC35C1 and VMA-21 by VMA21.
Response:
Thank you for noticing the inconsistencies, the spelling has been corrected.
* Figure 5:
Panel A, on the y-axis, for homogeneity reasons, I suggest to replace SCL35C1 by SLC35C1, SCL35A2 by SLC35A2 and VMA-21 by VMA21.
Response:
Thank you for noticing, the figure has been corrected.
* Supplementary figure S2:
Panel A, on the y-axis, for homogeneity reasons, I suggest to replace SCL35C1 by SLC35C1, and VMA-21 by VMA21.
Response:
Thank you for noticing, the figure has been corrected.
Tables
* Table S3
For homogeneity reasons, I suggest to replace VMA-21 by VMA21 in columns V and W
Response:
Thank you, the Table S3 is corrected.
* Table S5
For homogeneity reasons, I suggest to replace VMA-21 by VMA21 in columns V, W, AZ, BA, CD, CE, DH, DI, EL and EM
Response:
Thank you, the mistakes are corrected in Table S5.
References
* reference 7 → I recommend to check the name of the journal. From pubmed, I found “Japanese journal of human genetics”
* reference 18 → initials are missing for the last authors (Radboud J. E. M. Dolhain)
* reference 19 → one initial is missing for Kammeijer (Guinevere S. M. Kammeijer)
* reference 20 → one initial is missing for two of the authors: Vreeker (Gerda C.M. Vreeker) and van der Burgt (Yuri E.M. van der Burgt)
* reference 21 → initials are missing for Tollenaar (Rob A. E. M. Tollenaar)
* reference 22 → initials are missing for three of the authors: Selman (Maurice H.J. Selman), Hazes (Johanna M.W. Hazes) and Dolhain (Radboud J.E.M. Dolhain)
* reference 35 → correct the journal reference
eLife 2017;6:e22693, doi: 10.7554/eLife.22693
* reference 43 → correct the journal reference
Biology (Basel) 2017, 6, 16, doi:10.3390/biology6010016
* reference 46 → one initial is missing for two of the authors: de Brouwer (Arjan P M de Brouwer) and Deen (Peter M T Deen)
* reference 55 → correct the name of the journal: Front Immunol instead of Frontiers in immunology. Other Frontiers papers display the "short" name
* reference 61 → one initial is missing for Koeleman (Carolien A M Koeleman)
* reference 64 → one initial is missing for the first author (Maurice H J Selman)
Response:
Thank you for finding these inconsistencies, ref 7, 18, 19, 20, 21, 22, 35, 43, 46, 55, 61 and 64 have been corrected.
Reviewer 2 Report
The article investigates the N-glycans composition of IgG, transferrin and total plasma glycoproteins in patients affected by a small panel of CDGs. The purpose is very interesting and novel, although restricted to very small numbers. CDGs are so heterogeneous and in many cases so rare diseases that this is in part unavoidable. On the other hand, the approach is very complex and requires a lot of work. The study is performed rigorously and is technically sound.
There are some defects in the presentation that should be improved taking into account that IJMS has a wide audience that includes a large number of non-glycobiologists. In particular:
In the introduction, the authors should explain why they started from the selected CDG cases and the chosen glycoproteins.
They should add a table summarizing the main clinal features of each patient highlighting similarity and differences.
In the discussion, they should try to make a relationship between their findings and the clinical features.
Author Response
The article investigates the N-glycans composition of IgG, transferrin and total plasma glycoproteins in patients affected by a small panel of CDGs. The purpose is very interesting and novel, although restricted to very small numbers. CDGs are so heterogeneous and in many cases so rare diseases that this is in part unavoidable. On the other hand, the approach is very complex and requires a lot of work. The study is performed rigorously and is technically sound.
There are some defects in the presentation that should be improved taking into account that IJMS has a wide audience that includes a large number of non-glycobiologists. In particular:
In the introduction, the authors should explain why they started from the selected CDG cases and the chosen glycoproteins.
They should add a table summarizing the main clinal features of each patient highlighting similarity and differences.
In the discussion, they should try to make a relationship between their findings and the clinical features.
Response:
Thank you for your suggestions. We addressed the motivation of the chosen proteins in the introduction and we have included a column in Table 1 summarizing the main clinical features and manifestations of the various patients. Furthermore, the low number of CDG patients in the various subtypes precluded the association of specific glycan signatures with specific disease symptoms. Follow-up studies will be necessary, ideally with longitudinal monitoring of patients’ glycomic features, possibly in combination with therapy monitoring, to pinpoint towards specific associations of glycan signatures with (severity of) disease symptoms.